# Nanoclay and Polystyrene Type Efficiency on the Development of Polystyrene/Montmorillonite/Oregano Oil Antioxidant Active Packaging Nanocomposite Films

Aris E. Giannakas [1,*][iD], Constantinos E. Salmas [2,*][iD], Andreas Karydis-Messinis [2], Dimitrios Moschovas [2][iD], Eleni Kollia [3], Vasiliki Tsigkou [3], Charalampos Proestos [3][iD], Apostolos Avgeropoulos [2][iD] and Nikolaos E. Zafeiropoulos [2]

1   Department of Food Science and Technology, University of Patras, 30100 Agrinio, Greece
2   Department of Material Science & Engineering, University of Ioannina, 45110 Ioannina, Greece; karydis.and@gmail.com (A.K.-M.); dmoschov@uoi.gr (D.M.); aavger@uoi.gr (A.A.); nzafirop@uoi.gr (N.E.Z.)
3   Laboratory of Food Chemistry, Department of Chemistry, National and Kapodistrian University of Athens Zografou, 15771 Athens, Greece; elenikollia@chem.uoa.gr (E.K.); vtsigkou@chem.uoa.gr (V.T.); harpro@chem.uoa.gr (C.P.)
*   Correspondence: agiannakas@upatras.gr (A.E.G.); ksalmas@uoi.gr (C.E.S.)

**Abstract:** Over the years, there has been an effort to extend food shelf life so as to reduce global food waste. The use of natural biodegradable materials in production procedures is more and more adopted nowadays in order to achieve cyclic economy targets and improve environmental and human health indexes. Active packaging is the latest trend for food preservation. In this work, polystyrene was mixed with natural NaMt, OrgNaMt montmorillonite, and oregano essential oil to develop a new packaging film. Strength, oxygen and water-vapour permeation, blending and homogeneity, and antimicrobial and antioxidant activity were measured as basic parameters for food packaging films characterization. Instruments such as a tensile measurement instrument, XRD, FTIR, DMA, OPA (Oxygen Permeation Analyzer), and other handmade devices were used. Results showed that polystyrene could be modified, improved, and exhibits food odour prevention characteristics in order to be used for applications on food active packaging. The material with the code name PS5OO@OrgMt qualified between the tested samples as the most promising material for food active packaging applications.

**Keywords:** oregano oil; antioxidant activity; polystyrene; nanoclay; active packaging

## 1. Introduction

In recent decades much effort has been devoted to the use of polymer nanocomposites in food packaging industrial products [1]. Nanocomposites are multiphase materials where at least one dimension of the phases smaller than 100 nm [2]. Nanocomposites exhibit improved optical, thermal, barrier, and mechanical performance compared to the corresponding characteristics of conventional composites.

Polystyrene (PS) is one of the most widely used thermoplastic polymers in various industrial applications, including food packaging. Due to its high stiffness, strength, durability, good thermal properties, low moisture absorption, transparency, light density, convenience of processing and moulding, and low cost, this polymer is widely used as a food packaging material [3–6].

Among various nanoparticles [7] which are used as reinforcements for polymer nanocomposites, montmorillonite (Mt) has been extensively used in industrial applications because it is a naturally abundant clay mineral and biodegradable material, and it has unique structure and properties. PS/clay nanocomposites have been widely studied over the last several decades [8,9]. According to researchers, two main synthesis routes compose the mixing process of the inorganic Mt nanoclay with the hydrophobic PS matrix. The

first is the modification of the Mt surface with organic surfactants [10,11] and the second is the hydrophilic modification of the PS surface with compatibilizers such as co-maleic anhydride [12,13].

In the last few years, nanoclays have been suggested as ideal nanocarriers for bioactive compounds such as essential oils (EOs) [14–16]. According to later research findings, the EOs have antioxidant, antimicrobial, and aromatic properties and are promising candidates to replace synthetic additives and materials in chemoactive packaging which could cause adverse health effects [17,18]. Such novel EO@clay nanostructures can be easily loaded to polymer [19,20] or biopolymer [21–24] matrixes in order to develop a novel and promising active food packaging film. Such films are expected to exhibit enhanced mechanical and barrier properties due to the presence of the nanoclay. Moreover, the presence of the EOs is expected to provide aroma properties and controllable antioxidant activity [19,25–27]. Recently, a Na-montmorillonite clay (NaMt) and a commercial organically modified montmorillonite clay (OrgMt) were modified with thyme oil (TO), Oregano oil (OO), and Basil oil (BO) and used in a process for the development of low-density polyethylene active packaging films [19]. One of the results of this study was that even after six months of incubation, both EO@NaMt-based and EO@OrgMt-based films exhibited enhanced water/oxygen barrier properties and controllable and sustained antioxidant activity.

In this work, the OO@NaMt and OO@OrgMt hybrid nanostructures, which in a previous work exhibited the highest antioxidant activity [19], were loaded in polystyrene (PS) and polystyrene co-maleic anhydride (PS-coMA) matrixes for the development of active flavouring packaging films via a co-extrusion moulding process. The obtained PSOO@NaMt, PSco-mAOO@NaMt, and PSOO@OrgMt nanocomposite films were morphologically characterized via XRD analysis and FTIR spectrometry. The packaging performance of obtained films was evaluated via tensile properties, thermomechanical (DMA) studies, water, and oxygen barrier properties, antioxidant, and antimicrobial activity tests. Specific goals of this work are the efficiency of nanoclay type and polystyrene type in the development of polystyrene/montmorillonite/oregano oil active antioxidant/flavouring packaging films.

## 2. Materials and Methods

### 2.1. Materials

#### 2.1.1. Essential Oil Used

Origanum, EO was purchased from Esperis spa., Via A. Binda, 29, 20143 Milano (Italia). According to safety data sheets, the % mass composition of oregano oil was 60–70% carvacrol, 10–12.5% thymol, 10–12.5% paracymene, 5–7% alpha-pinene, 5–7% 1-Isopropyl-4-methyl−1,4-cyclohexadiene p-Mentha−1,4-diene, and 1–3% terpinene-4-olo, beta-myrcene and (R)-p-mentha−1,8-diene.

#### 2.1.2. Clay Used

Sodium exchanged montmorillonite (NaMt) (code name Nanomer® PGV) with mass density 2.6 g/cm³ and CEC value 145 meq/100 g was produced by Nanocor Company (Hoffman Estates, IL, USA) and supplied by Aldrich (St. Louis, MO, USA). The chemical composition of NaMt was 62.9% $SiO_2$, 19.6% $Al_2O_3$, 3.35% $Fe_2O_3$, 3.05% MgO, 1.68% CaO, 1.53% $Na_2O$. Organo-montmorillonite (OrgMt) NANOMER®-I·44P was produced by Nanocor Company (Hoffman Estates, IL, USA), and supplied by Aldrich (St. Louis, MO, USA). NANOMER®-I·44P is an -onium ion modified clay containing ~40 %wt. dimethyl dialkyl (C14−18) ammonium organic modifier.

#### 2.1.3. Polystyrene Used

Polystyrene (PS) was purchased from Aldrich Chemical Company (St. Louis, MO, USA) with weight and number average molecular weights of $M_w$ = 230,000 g/mol and $M_w$ = 140,000 g/mol, respectively. Polystyrene co-maleic anhydride (PS-coMA) was produced by Aldrich Chemical Company.

*2.2. Methods*

2.2.1. Preparation of OO@NaMt and OO@OrgMt Nanostructures

Both OO@NaMt and OO@OrgMt nanostructures were prepared via a green evaporation/adsorption process which was described in detail in previous work [14]. The obtained OO loading on OO@NaMt and TO@OrgMt was estimated gravimetrically, and it was approx. 24.1 %wt. and 36.0 %wt., correspondingly.

2.2.2. Preparation of PS/OO@NaMt, PSOO@OrgMt, and PScoMA/OO@NaMt Films

The PSOO@NaMt, PSOO@OrgMt, and PScoMA/OO@NaMt films (see Figure 1) were prepared via a melt-mixing process. For the preparation, a minilab twin extruder co-rotating (Haake Mini Lab II, ThermoScientific, ANTISEL, S.A., Athens, Greece) was used. The uniform operating temperature was 190 °C at a screw speed of 100 rpm for 5 min total processing time. The nominal compositions of NaMtEO and OrgMtEO nanohybrids which were added to PS and PScoMA were fixed to 3 %wt. and 5 %wt. The obtained melt compound strands were cut into small granules with a granulating machine. Finally, films were produced with approximately 10 cm diameter by hot-pressing of approximately 1.5 g of obtained granules at 160 °C under 3.0 megapascal (MPa) constant pressure for 3 min, using a hydraulic press with heated platens. "Blank" samples were also prepared by mixing PS with commercial NaMt and OrgMt and used for comparison measurements. Table 1 also lists the produced films used for comparison. It also lists the code names of all samples and the used amounts of PS, PScoMA, NaMt, OrgMt, OO@NaMt, and OO@OrgMt.

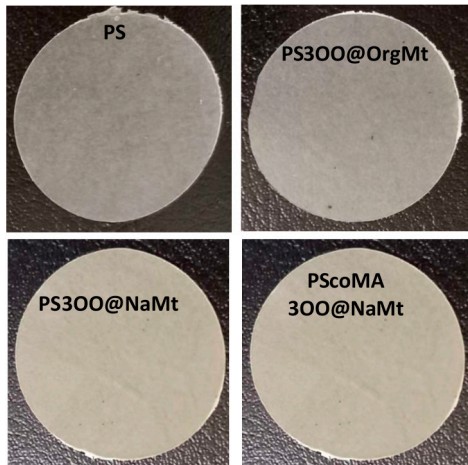

**Figure 1.** Representative photos of pure PS film and PS3OO@OrgMt, PS3OO@NaMt and PScoMA3OO@NaMt nanocomposite films.

**Table 1.** Code names and amounts used for all PSOO@NaMt, PSOO@OrgMt, and PScoMAOO@NaMt films.

|  | PS g | PScoMA g | NaMt g-%wt. | OrgMt g-%wt. | OO@NaMt g-%wt. | OO@OrgMt g-%wt. |
|---|---|---|---|---|---|---|
| PS | 5 | - | - | - | - | - |
| PS3NaMt | 4.85 | - | 0.15–3 | - | - | - |
| PS3OO@NaMt | 4.85 | - | - | - | 0.15–3 | - |
| PS5OO@NaMt | 4.75 | - | - | - | 0.25–5 | - |
| PScoMa | - | 5 | - | - | - | - |
| PScoMA3OO@NaMt | - | 4.85 | - | - | 0.15–3 | |
| PScoMA5OO@NaMt | - | 4.75 | - | - | 0.25–5 | |
| PS3OrgMt | 4.85 | - | - | 0.15–3 | - | - |
| PS3OO@OrgMt | 4.85 | - | - | - | - | 0.15–3 |
| PS 5OO@OrgMt | 4.75 | - | - | - | - | 0.25–5 |

### 2.3. XRD Analysis

The morphological characterization of all the obtained films was carried out via XRD measurements using a Brüker D8 Advance X-ray diffractometer (Bruker, Analytical Instruments, S.A. Athens, Greece) equipped with a LINXEYE XE High-Resolution Energy-Dispersive detector.

### 2.4. FTIR Spectrometry

The chemical structure of all the obtained films was confirmed by IR spectra measurements. Infrared (FTIR) spectra, which were the average of 32 scans at 2 cm$^{-1}$ resolution, measured with an FT/IR-6000 JASCO Fourier transform spectrometer (JASCO, Interlab, S.A., Athens, Greece) in the frequency range 4000–400 cm$^{-1}$.

### 2.5. Tensile Properties

Tensile measurements were carried out on all prepared films, according to the ASTM D638 method. A Simantzü AX-G 5kNt instrument (Simantzu. Asteriadis, S.A., Athens, Greece) was used. Three to five samples of each film were tensioned at an across head speed of 2 mm/min. The samples were dumbbell-shaped with gauge dimensions of 10 mm $\times$ 3 mm $\times$ 0.22 mm. Force (N) and deformation (mm) were recorded during the test.

### 2.6. DMA

The dynamic mechanical response was studied using a dynamic mechanical analyzer (DMA Q800, TA Instruments) in film tension mode. A temperature ramp of 5 °C/min from 30 °C to 120 °C, and a frequency of 1 Hz was applied in order to determine the storage modulus (E') and the loss factor (tan δ).

### 2.7. Water Vapor Transmission Rate (WVTR)

WVTR of all the obtained films was determined. Experimental conditions fixed at 38 °C and 50% RH according to the ASTM E96/E 96M-05 method. The used handmade apparatus and the applied procedure were described extensively in a previous publication [10]. For such measurements, film disks of 2.5 cm diameter and 100 μm thickness were used.

### 2.8. Oxygen Permeability (Pe$_{O2}$)

The oxygen transition rate (OTR) of all the obtained films was analyzed using an oxygen permeation analyzer (8001, Systech Illinois Instruments Co., Johnsburg, IL, USA). All samples were tested at 23 °C and 0% RH according to the ASTM D 3985 method. OTR values were expressed at cc $O_2$/m$^2$/day. The tested samples' OP values were calculated by multiplying the OTR values with the average film thickness of around 350–400 μm. The mean OTR value for each kind of film resulted from the measurements of three samples.

### 2.9. Antioxidant Activity

The antioxidant activity of films was evaluated using 300 mg of small pieces (approximately 3 mm $\times$ 3 mm) of each film. The sample was placed in a dark-coloured glass bottle with a plastic screw cap and filled with 10 mL of 50 ppm (mg/L) DPPH ethanolic solution. After incubation at 25 °C for 24 h in darkness, the % antioxidant activity values of the films were calculated according to the Equation (1):

$$\% \ Antioxidant \ activity = \frac{Abs_{control} - Abs_{sample}}{Abs_{control}} \times 100 \qquad (1)$$

### 2.10. Antimicrobial Activity Assay

The antimicrobial activity of films was examined by the agar diffusion method against Gram-negative bacteria Escherichia coli (ATCC 25922), Salmonella enterica subsp. enterica (DSMZ 17420), and Gram-positive bacteria Staphylococcus aureus (DSMZ 12463), Listeria

monocytogenes (DSMZ 27575). The bacteria species were provided from the Institute of Technology of Agricultural Products, ELGO-DEMETER, Lykovryssi, Greece.

Bacteria colonies were diluted in Mueller-Hinton broth and cultured overnight to obtain a range of 107–108 CFU mL$^{-1}$. The 24 h old cultures of bacteria were swabbed on Mueller-Hinton agar plates by rotating the plate every 60° to ensure homogeneous growth. Films were cut into a disc form of 6mm diameter using a circular knife and sterilized by a UV lamp. Film cuts were placed on Mueller-Hinton inoculated plates and incubated at 37 °C for 24 h. The diameter of inhibitory zones, as well as the contact area of the films with agar surface, was measured. The antibacterial activity of the OE used was also evaluated. The EO was cast into Mueller Hinton agar wells with 6 mm diameter and the clear zone of inhibition was recorded. The experiment was repeated thrice.

## 3. Results

### 3.1. XRD Results

Figure 2 shows the XRD patterns of all the obtained PSOO@NaMt, (Figure 2a) PScoMA/OO@NaMt, (Figure 2b), and PS/OO@OrgMt (Figure 1c) films. The XRD patterns of NaMt and OrgMt as received nanoclays, as well as of OO@NaMt and OO@OrgMt modified nanostructures are also included in this figure for comparison reasons. It is obvious from Figure 2a that the major diffraction peak of NaMt clay is located at around 2θ = 7.3° which corresponds to a d-spacing value of 1.21 nm. The OO@NaMt modified nanostructure exhibits a broader peak at 2θ = 6.8°.

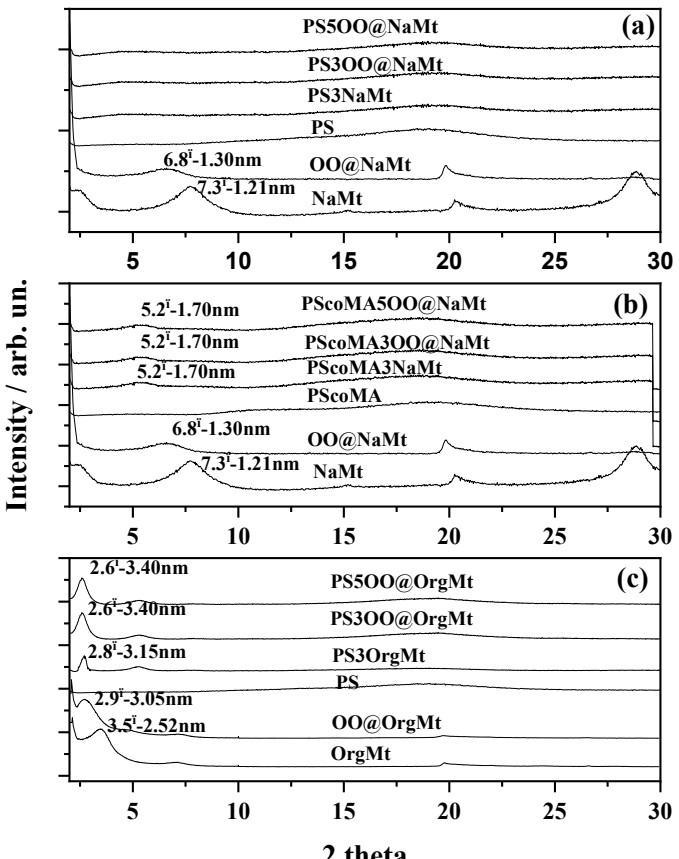

**Figure 2.** XRD plots of all PS/OO@NaMt, PSOO@OrgMt, PScoMA/OO@NaMt (**a**) films, OO@NaMt, OO@OrgMt nanostructures and, NaMt (**b**), and OrgMt clays (**c**).

According to a previous report [14], the modification of NaMt with EO molecules took place in the external surface of the NaMt layers and resulted in a partial exfoliation of clay platelets. XRD plots of the NaMt clay and OO@NaMt nanostructure blended with

PS chains exhibited no basal peak in the 2 theta range of 2 to 6° (see Figure 2a). Thus, it seems that PS chains cannot intercalate in the hydrophilic interlayer space of NaMt, which is something expected. The same happens in the case of OO@NaMt clay platelets where the modification of the NaMt with OO leads to the formation of a partially exfoliated nanocomposite structure which does not help the PS to further intercalate the clay. This result agrees with others in literature [28] where PS/OO@NaMt nanocomposites were prepared via an in situ polymerization method.

XRD plots (see Figure 2b) of PScoMANaMt and PScoMAOO@NaMt films exhibit a broad basal space peak at around $2\theta = 5.2°$ which corresponds to d-spacing value of 1.70 nm. This broad peak indicates that hydrophilic modified PScoMA chains intercalated partially the NaMt and OO@NaMt clays platelets. In Figure 1c the basal space of OrgMt clay is located at around $2\theta = 3.5°$ which corresponds to a d-spacing value of 2.52 nm. In the case of OO@OrgMt nanostructure, a clear shift of basal space at $2\theta = 2.9°$ value is observed. This observation is consistent with the results of a previous work [14] where it was reported that in the case of the modification of OrgMt clays with EO, the EO molecules intercalate the interlayer space of OrgMt platelets. Basal space of PS3OrgMt, PS3OO@OrgMt, and PS5OO@OrgMt films are located at 2theta values of 2.8°, 2.6°, and 2.6°, correspondingly. These values result in an increase of d-spacing at 3.15 nm for PS3OrgMt and at 3.40 nm for both PS3OO@OrgMt and PS5OO@OrgMt films. Moreover, indicate that PS was further intercalated the OO@OrgMt nanostructure, and a final secondary intercalated nanostructure is obtained. This result agrees with previous publications where the OO@OrgMt nanostructures favour the formation of nanocomposite structures intercalated with LDPE [19] and PS [28] matrixes.

### 3.2. FTIR Results

Figure 3 illustrates the FTIR scanning in the wavenumber region of 4000–400 $cm^{-1}$ for PS/clays nanocomposite films. The characteristic peaks around 3030 and 2930 $cm^{-1}$ of the PS spectra (Figure 3a–c), indicate the stretching of aromatic C-H and stretching vibration of aliphatic, respectively. The bands at 1599, 1496, and 1449 $cm^{-1}$ correspond to C=C stretching vibration, the band at 1368 $cm^{-1}$ indicates the vibrational mode of the $CH_2$ group, and the band at 698 $cm^{-1}$ is identified as the C-H benzenic stretching vibration. The broadband at 540 $cm^{-1}$ corresponds to the out of plane deformation of the phenyl ring [29]. For the PScoMA based films (see Figure 3b) the main differences in vibration bands compared to these of the PS films were caused by the presence of carbonyl group (C=O) of the MA and located at 3630 $cm^{-1}$, 1858 $cm^{-1}$, and 1779 $cm^{-1}$ [30].

In such PS/clay nanocomposites the interaction with clay platelets is indicated by the new bands at 3630, 918, 660, and 470 $cm^{-1}$. These bands are accredited to O-H stretching of structural hydroxyl groups, Al-O stretching, Mg-O bond, and Si-O stretching of the tetrahedral silica layers of modified clays, respectively [29,31]. In our case both the PSNaMt and the PScoMANaMt based films (Figure 2a,b, respectively) exhibit only the band at 470$cm^{-1}$ which corresponds to Si-O stretching. In Figure 2c the PSOrgMt films show the bands at 3630, and 470 $cm^{-1}$ which correspond to Al-O and Si-O stretching. This indicates a higher interaction between OrgMt based films and PS matrix compared to the interaction between NaMt based films and PS matrix and compared to the interaction between NaMt based films and PScoMA matrix.

### 3.3. Tensile Properties

Young's Modulus (E), tensile strength ($\sigma_{uts}$), and elongation at break ($\varepsilon_b$) are the mechanical properties of the tested films which are presented in Table 2.

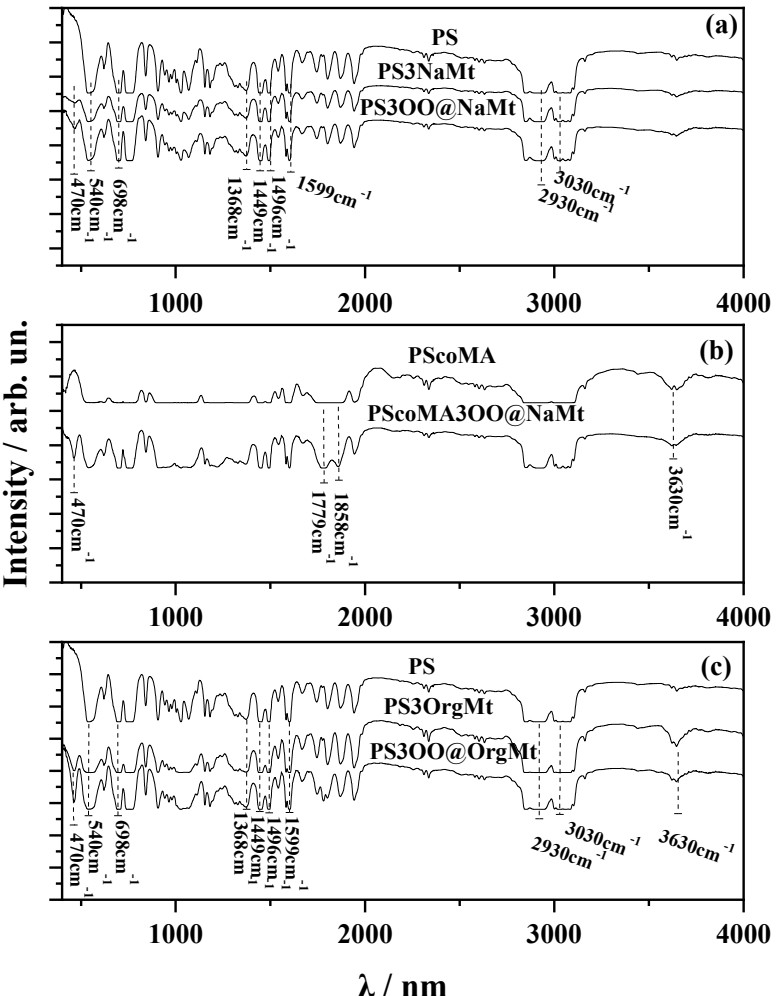

**Figure 3.** FTIR plots of all PS/ NaMt (**a**), PScoMANaMt (**b**), and PSOrgMt (**c**) films.

**Table 2.** Young's Modulus (E), tensile strength (σ_uts), and elongation at break (ε_b) values of all tested films.

|  | Young's Modulus-E (Mean.Dev) (N/mm$^2$) | σ (Mean.Dev) (N/mm$^2$) | %ε (Mean.Dev) |
|---|---|---|---|
| PS | 1832.7 (44.1) | 31.2 (2.0) | 1.9 (0.1) |
| PS3NaMt | 1740.2 (75.1) | 25.9 (2.5) | 0.8 (0.2) |
| PS3OO@NaMt | 1620.5 (69.4) | 23.9 (2.3) | 1.1 (0.1) |
| PS5OO@NaMt | 1590.5 (71.0) | 22.7 (3.3) | 1.0 (0.2) |
| PScoMa | 1815.7 (36.7) | 25.6 (4.4) | 1.5 (0.1) |
| PScoMA3OO@NaMt | 2270.5 (66.1) | 34.5 (4.1) | 1.3 (0.2) |
| PScoMA5OO@NaMt | 2380.8 (43.4) | 35.9 (4.0) | 1.3 (0.2) |
| PS3OrgMt | 1960.4 (61.2) | 32.8 (2.9) | 1.3 (0.2) |
| PS3OO@OrgMt | 2056.3 (58.8) | 33.3 (3.6) | 1.4 (0.2) |
| PS 5OO@OrgMt | 2120.5 (53.1) | 33.6 (3.9) | 1.5 (0.2) |

As for Young's Modulus (E), it is obvious from the measurements that the addition of maleic anhydride in PS does not affect this property. The addition of NaMt in the PS matrix causes a decrease of (E) values while the addition of OrgMt causes an increase. This behaviour is enhanced by the addition of OO in both cases i.e., OO@NaMt and OO@OrgMt. However, when the polymeric matrix changes to PScoMa the addition of OO@NaMt change behaviour and causes a high increase of the (E) values.

As for the tensile strength (σ), the addition of coMa in PS causes a decrease in this property. In all the other cases the behaviour of this mechanical property is identical to the behaviour of the (E).

As for the elongation at break values, it seems that the addition of NaMt decreases values more than the decrease causes the addition of OrgMt which, however, is higher than the decrease causes the coMa. The addition of OO seems to operate in an opposite way, increasing the elongation at break values.

Figure 4 depicts a direct assessment of the stress-strain behaviour of the "after growth" films. It is obvious from this figure that the highest increase in stiffness and strength values are obtained for PScoMA5OO@NaMt and PS5OO@OrgMt films. According to the XRD results an exfoliated structure was indicated for $PS_xOO@NaMt$ samples, a partial intercalated structure for $PScoMa_xOO@NaMt$ samples, and an intercalated structure for $PS_xOO@OrgMt$ samples. These results are following the tensile properties variation.

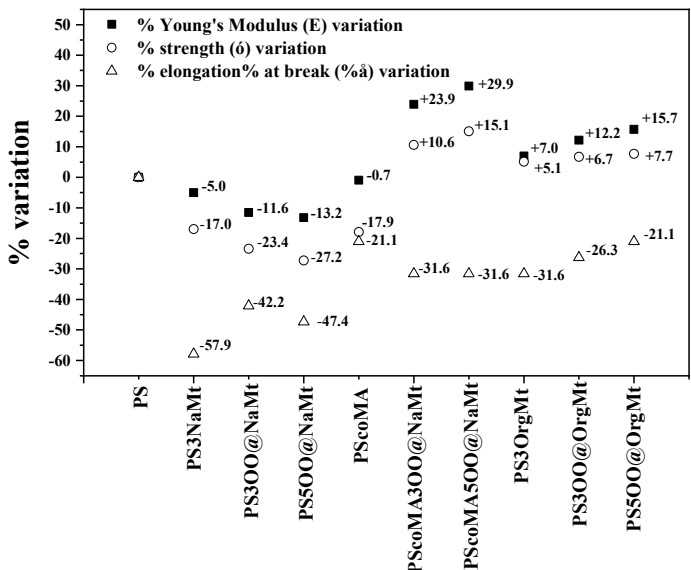

**Figure 4.** %Variation of Young's Modulus-E values, tensile strength (σ) values, and %elongation at the break-%ε values for all tested PS based active packaging films.

### 3.4. DMA Results

The storage modulus and tan δ of pure PS, PS3NaMt and PS3OO@NaMt are shown in Figure 5a,d, respectively. The $T_g$, as well as the storage modulus (at 40 °C and 100 °C), are shown in Table 3. The $T_g$ values were calculated from the peak of tan δ. The storage modulus of the samples PS3NaMt and PS3OO@NaMt were found to be higher than that of pure PS for the whole temperature range of the experiment. The higher storage modulus of PS3NaMt could be attributed to the stiffening effect of NaMt [32]. Our results are in good agreement with previous reports [32,33]. The values of the Tg slightly increased with the addition of clay while the addition of oregano oil led to a decreased Tg, implying a plasticizing action [34]. The increased storage modulus observed after the addition of oregano oil has not been reported for the present system in the literature, but a similar behaviour has been reported in a different system consisting of whey protein isolate films incorporated with oregano oil [34]. In Figure 5b,e, storage modulus and tan δ curves of PScoMA, PScoMa3OO@NaMt and PScoMa5OO@NaMt are shown. The storage modulus increased with the addition of clays. The highest storage modulus was obtained when 3 %wt. NaMtOO was added to the PScoMA matrix. The $T_g$ of the nanocomposite PScoMa5OO@NaMt was increased by 4 °C compared to PScoMA while the $T_g$ of the PScoMa3OO@NaMt was slightly decreased. The values are shown in Table 3.

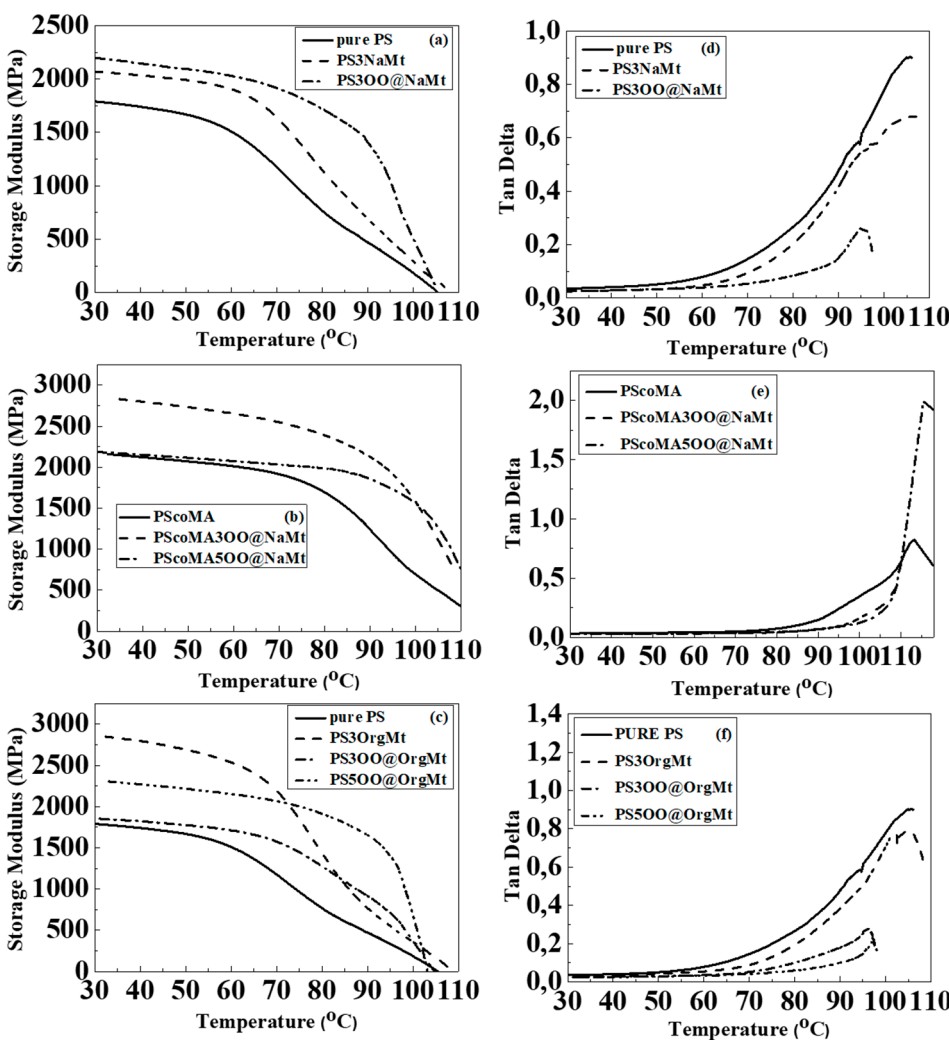

**Figure 5.** Left part: Storage modulus of (**a**) pure PS, PS3NaMt and PS3OO@NaMt, (**b**) PScoMA, PScoMA3OO@NaMt, PScoMA5OO@NaMt (**c**) pure PS, PS3OrgMt, PS3OO@OrgMt and PS5OO@OrgMt. Right part: Values of ta δ of (**d**) pure PS, PS3NaMt and PS3OO@NaMt, (**e**) PScoMA, PScoMA3OO@NaMt, PScoMA5OO@NaMt and (**f**) PS, PS3OrgMt, PS3OO@OrgMt and PS5OO@OrgMt.

**Table 3.** Storage modulus (at 40 °C and 100 °C) and $T_g$ values of all the tested films.

| Sample | E′ (40 °C) | E′ (100 °C) | Glass Transition Temperature ($T_g$) |
|---|---|---|---|
| PS | 1739 MPa | 172 MPa | 105 °C |
| PS3NaMt | 2034 MPa | 280 MPa | 107 °C |
| PS3OO@NaMt | 2146 MPa | 510 MPa | 96 °C |
| PScoMA | 2121 MPa | 689 MPa | 112 °C |
| PScoMA3OO@NaMt | 2799 MPa | 1542 MPa | 110 °C |
| PScoMA5OO@NaMt | 2151 MPa | 1556 MPa | 116 °C |
| PS3OrgMt | 2795 MPa | 356 MPa | 104 °C |
| PS3OO@OrgMt | 1823 MPa | 356 MPa | 96 °C |
| PS5OO@OrgMt | 2271MPa | 683 MPa | 97 °C |

In Figure 5c,f, storage modulus and tan δ curves of PS3OrgMt, PS3OO@OrgMt and PS5OO@OrgMt compared to the values of pure PS can be seen. The addition of clay increased the storage modulus while the $T_g$ slightly decreased, which is in good agreement with the results reported by Fu and Qutubuddin [35]. The same research group showed that as the clay content increased the polystyrene molecular weight decreased [35]. This

may be a possible explanation for the slight decrease in Tg that was observed in PS3OrgMt compared to pure PS. After the examination of PS3OO@OrgMt and PS5OO@OrgMt storage modulus curves, in a temperature range (85–100 °C and 75–110 °C, respectively) the storage modulus increased. This phenomenon has not been reported in the literature for such materials to the best of our knowledge. The modulus enhancement through antiplasticization has been explained by Mascia et al. [36–39] for plasticization of PVC with tricresyl phosphate. According to Mascia et al. [39], this behaviour may be attributed to morphological changes caused by the plasticizer property to induce phase separation through internal associations driven by crystallization or even vitrification, albeit in the present system the crystallization mechanism is not relevant. The addition of oregano oil led to a decrease in $T_g$ because of the plasticization effect [34].

### 3.5. Barrier Properties

Table 4 lists the obtained OTR and WVTR values for the tested films. Multiplying the obtained OTR values (cc STP/m²/day) with the average film thickness (m) the Oxygen Permeability (Pe$_{O2}$ cc STP/m/day) values were calculated and presented. Assuming that the silica absorber in the WVTR handmade device absorbs 100% of the permeated humidity, and knowing that according to the ASTM E96/E 96M-05 method in the opposite side of the examined film the RH is 50 % (i.e., 22.86747 g/cm³), we could assume that the concentration difference ΔC between the two sides of the film is ΔC = 22.86747 g/cm³. Solving Fick's 1st law to the diffusion coefficient D (cm²/s) we see that we can calculate this parameter by multiplying the measured WVTR (g/s) average values with the average film thickness Δx (cm) and divide by the difference of concentration ΔC (g/cm³) and by the cross-sectional area of the film A = 4.91 cm². Results are presented in Table 4.

**Table 4.** Average films thickness, Water Vapor Transmission Rate (WVTR), water sorption, obtained Oxygen Transmission Rate (O.T.R.), Oxygen Permeability (Pe$_{O2}$), and Antioxidant Activity values of all tested films.

| Code Name | Aver. Film Thick. (mm) | W.V.T.R ×10⁶ (g/s) (Mean.Dev) | D$_W$ × 10¹⁰ (cm²/s) | Aver. Film Thick. (mm) | O.T.R. (cc/m²/Day) (Mean.Dev) | Pe$_{O2}$ × 10³ (cc/m/Day) (Mean.Dev) | Antiox. Activ. after 24 h |
|---|---|---|---|---|---|---|---|
| PS | 0.1 | 1.3704 (0.02) | 1.221 | 0.375 | 1715.0 (13.2) | 643.1 (4.4) | n.d. |
| PS3NaMt | 0.1 | 1.5000 (0.02) | 1.336 | 0.375 | 1934.0 (15.8) | 725.2 (5.3) | n.d. |
| PS3OO@NaMt | 0.1 | 1.4593 (0.03) | 1.300 | 0.375 | 1885.0 (15.6) | 706.9 (5.2) | 28.1 (2.2) |
| PS5OO@NaMt | 0.1 | 1.4482 (0.03) | 1.290 | 0.375 | 1932.0 (15.4) | 724.5 (5.1) | 31.2 (1.9) |
| PScoMa | 0.1 | 1.3778 (0.02) | 1.227 | 0.375 | 1731.0 (13.2) | 649.1 (4.4) | n.d. |
| PScoMA3OO@NaMt | 0.1 | 1.3482 (0.02) | 1.201 | 0.375 | 1684.0 (13.0) | 631.5 (4.4) | 27.3 (2.0) |
| PScoMA5OO@NaMt | 0.1 | 1.2407 (0.01) | 1.105 | 0.375 | 1602.0 (12.8) | 600.8 (4.3) | 32.6 (1.7) |
| PS3OrgMt | 0.1 | 1.1630 (0.02) | 1.036 | 0.375 | 1325.0 (12.7) | 496.9 (4.2) | n.d. |
| PS3OO@OrgMt | 0.1 | 1.1222 (0.02) | 1.000 | 0.375 | 1245.0 (12.7) | 466.9 (4.2) | 37.2 (2.1) |
| PS 5OO@OrgMt | 0.1 | 1.0963 (0.01) | 0.977 | 0.375 | 1142.0 (12.6) | 428.3 (4.2) | 50.4 (1.8) |

It is obvious from these values that both water and oxygen permeability follow the same trend, i.e., both the water and oxygen barrier values of both the PSNaMt and PSxOO@NaMt films decreased while the same properties of both the PScoMAxOO@NaMt and PSxOO@OrgMt films increased. Water and oxygen barrier values trend is following the structural characterization results of films where an exfoliated, partial intercalated and strong intercalated structure was revealed for PSxOO@NaMt, PScoMAxOO@NaMt, and PSxOO@OrgMt films, respectively. The lowest water and oxygen permeability values are obtained for PS3OO@OrgMt and PS5OO@OrgMt samples. XRD results indicate an intercalated nanocomposite structure for both these samples.

### 3.6. Antioxidant Activity

Table 4 includes also the values of the antioxidant activity calculated using Equation (1). Significant antioxidant activity is exhibited by all the tested films. This indicates that all obtained films could be potentially used as active packaging films. More specifically, antioxidant activity increases as the content of OO@NaMt and OO@OrgMt nanostructure

increases. Higher antioxidant activity values were obtained for OO@OrgMt based films than for the OO@NaMt based films. As it was mentioned in the preparation section the highest OO content was calculated gravimetrically for OO@OrgMt nanostructure (i.e., 36.0%) than for OO@NaMt nanostructure (i.e., 20.1%).

### 3.7. Antimicrobial Activity

No formation zone around the enhanced film discs was observed indicating the lack of antibacterial activity. Moreover, films also did not show any antibacterial effect on the contact area under the discs. However, the EO used as a control sample exhibited strong inhibitory activity against the tested bacteria. Specifically, oregano oil showed total inhibition of *E. coli* and L. monocytogenes growth. Moreover, it presented also good antibacterial activity against *S. aureus* and *S. enterica* with a mean diameter of 65 mm and 30 mm of inhibition zone, respectively. The results demonstrated that pure EO has a clear effect on all tested bacteria. Since pure EO showed a significant antibacterial activity it was expected for the enhanced with EOs films to show some activity too.

However, the lack of antibacterial activity may be attributed to the film formation process by hot-pressing at a high temperature (190 °C) which could affect the bioactive components of the EO and lead to loss of volatile compounds. Therefore, higher concentrations of EO need to be incorporated into the films in order to achieve any antimicrobial effect.

Previously Kamel et al. (2021) found that polystyrene films incorporated with clove oil at concentration 10–30% did not exhibit any antimicrobial effect on the tested microorganisms (*E. coli* & *S. aureus*), while polystyrene films incorporated with a higher concentration of clove oil (40%) showed antibacterial activity with an inhibition zone of 4mm and 5.2 mm diameter, for *E. coli* and *S. aureus*, respectively. However, Kamel et al. (2021) did not involve high temperature during film formation; consequently, any reduction in the antibacterial efficacy was attributed to the loss of volatile compounds through the drying process [40]. Dias et al. [25] also developed flavouring active LDPE based lemon aroma films with no antimicrobial activity. This result suggests that all the obtained PSxOO@NaMt, PScoMAxOO@NaMt and PSxOO@OrgMt films could potentially be used rather as active aroma films to protect meat products from lipid oxidation and prevent meat odour rather than as antimicrobial packaging films.

### 3.8. Statistical Analysis of the Experimental Data

Three to five samples were measured for every kind of film to achieve a mean value for each one of the properties i.e E, $\sigma_{uts}$, %$\varepsilon$, WVTR, %water sorption, OP, and %Antioxidant activity after 24 h. Measurements were statistically interpreted using the statistical software SPSS ver. 20. A confidence interval C.I. = 95% which is the most common value used for such analyses was assumed for every test. Thus, the value of the statistical significance level was a = 0.05. Results for mean values and standard deviation of the above-mentioned parameters are presented in Table 2 and e.g., 4. Furthermore, hypothesis tests were carried out to support that considering different samples, every property has a statistically different mean value.

## 4. Conclusions

XRD analysis revealed that PS chains cannot intercalate the hydrophilic interlayer space of the NaMt, which was expected. The same happened with the partially exfoliated OO@NaMt nanocomposite structure. The modified PScoMa hydrophilic chains partially intercalated the NaMt and the OO@NaMt clays. In the case of OrgMt clays, the OO fully intercalated such clays and secondary intercalation took place when the PS or PScoMa matrixes intercalated the OO@OrgMt nanostructure.

FTIR measurements show that the higher interaction between clay and PS or PScoMa matrix existed in the cases of OrgMt compared to the cases of NaMt.

Tensile tests show that the mixing of OrgMt with the PS or PScoMa increased the Young Modulus (E) as well as the tensile stress (σ), while the addition of NaMt decreased these properties.

The elongation at break values decreased in both cases of clay addition, i.e., NaMt or OrgMt. Nevertheless, this could be balanced with the addition of OO which increases this property.

DMA tests show that the mixing of coMa, OO, NaMt, and OrgMt with PS matrixes increased the storage modulus in both temperatures of 40 °C and 100 °C. In all cases, the glass transition temperature remained in the region of 96–116 °C.

Oxygen and Water-Vapor barrier tests show that the samples which exhibited the lowest permeability were PS3OO@OrgMt and the PS5OO@OrgMt.

Antioxidant activity tests show that the OrgMt based films exhibited the higher antioxidant activity because of the higher OO content. This activity increased with the increase of this content.

Antimicrobial activity tests show that even though the pure OO exhibited strong antimicrobial activity, no significant antimicrobial effect was observed in all cases of PS/clay films.

The conclusion of this work is that the PScoMAxOO@NaMt and PSxOO@OrgMt films are promising candidate materials for future packaging applications as aroma active and/or antioxidant active films which could protect food products from lipid oxidation and prevent undesirable food odours.

**Author Contributions:** The synthesis experiments design, A.E.G. and C.E.S.; Characterization measurements carrying out and interpretation, A.E.G. and C.E.S.; Paper writing, A.E.G. and C.E.S.; The overall evaluation of this work, A.E.G. and C.E.S.; Experimental data analysis and interpretation, A.E.G., C.E.S., N.E.Z., A.A. and C.P.; The XRD, FTIR, OTR, tensile measurements, antioxidant activity, and WVTR experimental measurements carrying out, A.E.G. and C.E.S.; DMA experiment carrying out D.M. and A.K.-M.; Antimicrobial activity tests carrying out E.K. and V.T. All authors have read and agreed to the published version of the manuscript.

**Funding:** This research was funded by the funding programme "MEDICUS", of the University of Patras.

**Conflicts of Interest:** The authors declare no conflict of interest.

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
