# Peer review of "Nanoclay and Polystyrene Type Efficiency on the Development of Polystyrene/Montmorillonite/Oregano Oil Antioxidant Active Packaging Nanocomposite Films"

_applsci, doi:10.3390/app11209364_

Round 1
Reviewer 1 Report
The manuscript of Aris E. Giannakas et al deals with the preparation and mechanical investigation of PS-nanoclay composites in combination with oregano oil. It seems like a follow up of previous work, however it may contain enough novelty to be accepted in Applied Sciences after a major revision.
My major concerns:
The English of the manuscript needs to be corrected by a native speaker/lector. It is crowded with grammatical errors and mistyping.
The scientific notations are terrible. In many places there is no space between the number and the unit and the units are mistyped:
A few examples:
- In 2.1.2. Clay used
Please watch for the correct use of the sub- and superscripts in the units and chemical formulas, such as “g/cm3, Al2O3, Fe2O3”
- In 2.1.3.
Please correct the molecular weights “Mw¼230,000 g=mol and Mn¼140,000 g=mol” Mw=230,000 g/mol…
- Line 109 “to 3 wt.% and 5%wt.” Be consistent with the notation of the wt %!
Line 156, Eq. 1. The equation for % Antioxidant activity is wrong. If you multiply Abs(control) with Abs(sample) and divide by Abs(control) you will get Abs(sample) which in not correct.
The correct formula is (Abs(control)-Abs(sample))/Abs(control)*100
If the new composites were to be used as food packaging a few pictures of them needs to be included in the manuscript.
In Table 1 what does “gr” stand for? For example in “PS-gr”
Author Response
Dear Editor
Please find attached the revised version of our paper applsci-1407657_Revised, entitled “Nanoclay and polystyrene type efficiency on the development of polystyrene/montmorillonite/oregano oil antioxidant active packaging nanocomposite films”, authored by Aris E. Giannakas*, Constantinos E. Salmas*, Andreas Karydis-Messinis, Dimitrios Moschovas, Eleni Kollia, Vasiliki Tsigkou, Charalampos Proestos, Apostolos Avgeropoulos, Nikolaos E. Zafeiropoulos. to be published in: applied science journal, to section: Chemistry, and special issue: Antioxidants in Natural Products II.
All the reviewers’ queries have been answered.
All changes have been made according to the reviewers’ comments and highlighted in the text with red colored letters.
We are at your disposal for any further question.
Sincerely yours
Constantinos Salmas Aris Giannakas
Assistant Professor Assistant Professor
Department of Materials Sci. & Engineering Department of Food Science & Technology
University of Ioannina University of Patras
Reviewer#1 comments
The manuscript of Aris E. Giannakas et al deals with the preparation and mechanical investigation of PS-nanoclay composites in combination with oregano oil. It seems like a follow up of previous work, however it may contain enough novelty to be accepted in Applied Sciences after a major revision.
My major concerns:
- The English of the manuscript needs to be corrected by a native speaker/lector. It is crowded with grammatical errors and mistyping.
- The scientific notations are terrible. In many places there is no space between the number and the unit and the units are mistyped:
A few examples:
In 2.1.2. Clay used
Please watch for the correct use of the sub- and superscripts in the units and chemical formulas, such as “g/cm3, Al2O3, Fe2O3”
In 2.1.3.
Please correct the molecular weights “Mw¼230,000 g=mol and Mn¼140,000 g=mol” Mw=230,000 g/mol…
Line 109 “to 3 wt.% and 5%wt.” Be consistent with the notation of the wt %!
Line 156, Eq. 1. The equation for % Antioxidant activity is wrong. If you multiply Abs(control) with Abs(sample) and divide by Abs(control) you will get Abs(sample) which in not correct.
The correct formula is (Abs(control)-Abs(sample))/Abs(control)*100
If the new composites were to be used as food packaging a few pictures of them needs to be included in the manuscript.
In Table 1 what does “gr” stand for? For example in “PS-gr”
Authors reply
We thank reviewer#1 for his/her efforts to evaluate the current study. A point-by-point answer in all reviewer’s comment follows.
The English of the manuscript was now fully corrected by a native speaker/lector and all the revisions highlighted with red text.
All the issues about the scientific notations were corrected according to reviewer\s suggestions and highlighted with red text.
- All the issues for the correct use of the sub- and superscripts in the units and chemical formulas were corrected and highlighted with red text.
- Mw was corrected and highlighted with red text.
- The notation of the % wt. corrected all over the revised paper properly.
- 1 was corrected in the revised manuscript
- In Table 1 the gr was replaced by g which is the correct unit symbol of mass according to SI.
Reviewer#2 comments
The manuscript presents a quite interesting approach to developing a polystyrene/montmorillonite/oregano oil nanocomposite film which could be applied as an antioxidant packaging material. The description of the examination of the structure and properties of the developed material does not raise any objections. The only remark concerns the formula used to calculate the antioxidant activity (line 156). Shouldn't the sum of sample and control absorbance be included in the numerator instead of multiplying these values? Apart from this remark, the research was conducted in a required manner with due diligence. The presented research results confirm the properties of the developed material and refer to previous scientific reports.
Authors reply
We thank reviewer#2 for his/her positive comments and his/her efforts to evaluate of this study.
In the revised version the eq (1) with the calculation of antioxidant activity corrected:
% Antioxidant activity = (Abscontrol - Abssample)/Abscontrol) x 100 (1)

Reviewer 2 Report
The manuscript presents a quite interesting approach to developing a polystyrene/montmorillonite/oregano oil nanocomposite film which could be applied as an antioxidant packaging material. The description of the examination of the structure and properties of the developed material does not raise any objections. The only remark concerns the formula used to calculate the antioxidant activity (line 156). Shouldn't the sum of sample and control absorbance be included in the numerator instead of multiplying these values? Apart from this remark, the research was conducted in a required manner with due diligence. The presented research results confirm the properties of the developed material and refer to previous scientific reports.
Author Response

(The authors gave the same response as above.)

Round 2
Reviewer 1 Report
The manuscript can now be accepted.